Range expansion of a fouling species indirectly impacts local species interactions

Speights Cori J. cjs815@msstate.edu cori.speights@gmail.com 1 2
McCoy Michael W. 2
1 Department of Biological Sciences, Mississippi State University , Starkville , MS , United States of America
2 Department of Biology, East Carolina University , Greenville , NC , United States of America
Collins Tim
Electronic publication date: 2017 Oct 19
Publication date: 2017
Volume: 5
Electronic Location ID: e3911
Received 2017 Jun 30; Accepted 2017 Sep 19
Copyright: ©2017 Speights and McCoy
Copyright year: 2017
Copyright holder: Speights and McCoy
License: This is an open access article distributed under the terms of the Creative Commons Attribution License, which permits unrestricted use, distribution, reproduction and adaptation in any medium and for any purpose provided that it is properly attributed. For attribution, the original author(s), title, publication source (PeerJ) and either DOI or URL of the article must be cited.
License URL: https://creativecommons.org/licenses/by/4.0/

Keywords: Boring sponge, Climate change, Fouling, Functional response, Indirect interactions, Predation, Oyster

Funding: National Science Foundation Award #1556743 This research was supported by East Carolina University and in part by a National Science Foundation Award (#1556743) to Michael McCoy. There was no additional external funding received for this study. The funders had no role in study design, data collection and analysis, decision to publish, or preparation of the manuscript.

==============================
We investigated how recent changes in the distribution and abundance of a fouling organism affected the strength of interactions between a commercially important foundation species and a common predator. Increases in the abundance of boring sponges that bioerode the calcified shells of oysters and other shelled organisms have been attributed to increased salinization of estuarine ecosystems. We tested the hypothesis that fouling by boring sponges will change the interaction strength between oysters and a common predator (stone crabs). We generated five oyster density treatments crossed with two sponge treatments (sponge and no sponge). We contrasted the interaction strength between stone crabs and fouled and non-fouled oysters by comparing the parameters of fitted functional response curves based on Rogers random predation model. We found that fouled oysters suffered higher predation from stone crabs, and that the increased predation risk stemmed from a reduction in the handling time needed to consume the fouled oysters. These findings highlight the importance of understanding the effects of abiotic changes on both the composition of ecological communities, and on the strengths of direct and indirect interactions among species. Global climate change is altering local ecosystems in complex ways, and the success of restoration, management, and mitigation strategies for important species requires a better appreciation for how these effects cascade through ecosystems.

Introduction

The strength of interactions between predators and prey can be dependent upon ecological context and a plethora of environmental variables (Grabowski, 2004; Laudien & Wahl, 1999; Menge, 1995; Wahl, Hay & Enderlein, 1997). For example interactions with the abiotic environment (e.g., temperature, carbon dioxide, sea level rise) can change activity levels or physiological processes (Gilman et al., 2010), and the presence of other organisms can directly or indirectly change the strength of species interactions (Preisser, Bolnick & Benard, 2005; Werner & Peacor, 2003). For example, Schmitt, Osenberg & Bercovitch (1983) showed that drill holes on kelp snails caused by failed octopus predation attempts resulted in increased barnacle fouling of the snail’s shells. The increased fouling increased the chances of the snails being dislodged from kelp, increasing their exposure to benthic predators by increasing the amount of time spent on the bottom rather than on kelp. Fouling organisms on marine mollusks can also increase susceptibility to predators by compromising the integrity of protective shells (Duckworth & Peterson, 2013). These indirect effects, where one species alters the strength of interactions between other species, may become more common and important as species invasions or range expansions resulting from environmental change lead to novel direct and indirect species interactions (Gilman et al., 2010; Kordas, Harley & O’Connor, 2011; Walther, 2010).

In marine and estuarine ecosystems, increases in temperature, salinity, and dissolved pCO2 that are predicted to occur over the next several decades may decrease local habitat quality for some species while facilitating invasions and range expansions for others (Sorte, Williams & Carlton, 2010; Sunday et al., 2016). Understanding how changes in biotic and abiotic conditions of ecosystems may change species interactions might be particularly important for foundation species and the communities that depend on their biogenic habitat structures (Hoegh-Guldberg et al., 2007). For example, oysters are foundation species in estuaries because their biogenically formed calcium carbonate shells provide habitat structure and refuge that support many other species (Gutiérrez et al., 2003). Oysters also provide services such as water filtration that reduces eutrophication, and their reefs provide coastal protection (Meyer, Townsend & Thayer, 1997; Newell, 2004; Van Wesenbeeck et al., 2013). Therefore, changes in the distribution of predators or fouling species that affect the health or survival of oysters can have important implications for both oysters and oyster reef communities and the services they provide.

In this study we investigated how the interactions between oysters and a common oyster predator are influenced by a bioeroding sponge which may be expanding its distribution as a result of increasing salinity and temperature in coastal estuaries (Hong & Shen, 2012; Lindquist, 2011). Specifically, we investigated how the presence of boring sponges, Cliona spp., impact trophic interactions between eastern oysters, Crassostrea virginica, and an important native predator, the stone crab Menippe mercenaria. While studies have shown stone crabs can have less of an effect on oyster reefs than other mesopredators (e.g., mud crabs) they have recently increased establishment in North Carolina oyster reefs (Lindquist, 2011; Rindone & Eggleston, 2011), and we still do not know the magnitude of their effects on oysters interacting with other species, such as sponges. Boring sponges bioerode the calcium carbonate substrates on which they settle (Duckworth & Peterson, 2013; Fang et al., 2013). Mollusks that are hosts to boring sponges have weakened shells (Stefaniak, McAtee & Shulman, 2005), slower growth, reduced condition, and lower survival than mollusks lacking these bioeroding colonists (Carroll et al., 2015). Therefore, we quantified the effects of boring sponges on the interaction strength between stone crabs and fouled and non-fouled oysters. We compared the shape of the crab’s functional response to test the hypothesis that the weakened shells of fouled oysters caused by boring sponges will increase the strength of the predator–prey interaction. We focused on the functional response because it is the most direct measure of the interaction strength between predators and prey and it provides a mechanistic link to their population dynamics.

Specifically, to determine the effect of sponges on oyster survival we compared the parameters of type II functional responses (i.e., changes in attack rates or handling times). If sponges are distasteful then crabs will be more likely to avoid foraging on fouled oysters and this affect will be manifested in differences in attack rates. In contrast, if sponges cause changes in shell strength that facilitate crab predation, then we might expect to see shorter handling times and thus higher maximum consumption rates by crabs on fouled oysters.

Methods

Stone crabs were collected from Middle Marsh in Beaufort, North Carolina (NCDMF Permit No. 706671) and allowed to acclimate in 0.6 m2 tanks at the Duke Marine lab for at least 48 h. Each tank received a constant flow of unfiltered seawater and a piece of PVC pipe was provided for refuge. Ten crabs were each wet weighed (g) and the length (mm) of their carapace measured with digital calipers (mean ± st. error: 93.6 ± 10.9 g and 64.1 ± 3.1 mm, respectively). The stone crabs were maintained on a 12 h light/dark cycle and starved for 48 h prior to the beginning of the experiment. Oysters were collected around Morehead City, NC and sorted into two groups: fouled or non-fouled by boring sponge. Oysters of similar sizes (mean ± st. error: 5.07 ± 0.07 cm) were used to generate five oyster density treatments of 1,2,4,8, and 22 oysters crossed with two sponge treatments (sponge and no sponge). Oysters were added to each stone crab tank at noon on the day of the experiment. The temperature was recorded in an unused tank at the start of each trial (28.4, 27.7, 25.4, and 24.3 °C, for trials 1, 2, 3, and 4 respectively). The number of oysters eaten was recorded via visual surveys after 24 h and all remaining oysters were removed.

Following each trial, each crab was then fed two oysters a day for three days after which any non-consumed oysters were removed and the crabs were again starved for 48 h and re-randomized for use in another replicate. While not ideal, methods for reusing stone crabs through feeding standardization between trials has been previously reported (Wong, Peterson & Kay, 2010). In lieu of using new stone crabs for each trial as has been done in previous studies, by using the same crabs, each was influenced by the same background environment before each trial. Additionally, a previous study with rock crabs (Cancer irroratus) showed that reused crabs had no change in mussel capture behavior over a three month holding period (Matheson & Gagnon, 2012). To ensure that any uncertainty due to individuals differences among crabs were accounted for we randomly assigned each crab to a sponge × density treatment for each trial. This distributed any individual crab effect randomly across treatments which minimizes biases in model fits. Two crabs that never consumed oysters in the lab were replaced by new wild-caught crabs for trials. This experiment was replicated four times and oyster collections were made each week to ensure survival of oysters throughout the experiment.

Data were analyzed in the R statistical programming environment (R Core Team, 2016). Specifically, we fit a Type II functional response curve using Rogers’ random predation model (Juliano, 2001; Rogers, 1972) to quantify predation rates for oysters with and without boring sponge. We used Rogers’ formulation because it corrects for prey depletion that occurs as a result of predation over the course of the experiment. The random predator model predicts the number of prey eaten, N, as: (1) N=N01−e−aT−hN

where T is time, N0 is the initial prey abundance, h is time spent handling prey and a is the instantaneous attack rate. Rogers’ equation can be solved iteratively (Juliano, 2001) as expressed in Eq. (1), however we fit our data to a closed-form solution by expressing Eq. (1) in terms of Lambert’s W function (Bolker, 2008; McCoy & Bolker, 2008) so that the number of prey eaten, N, equals: (2) N=N0−WahN0e−aT−hN0ah.

Models were fit using the method of maximum likelihood in the bbmle package (Bolker & R Development Core Team, 2016) with a binomial error distribution. Specifically, we used a flexible parameter approach to fit (1) a model that estimated attack rates and handling times for the two prey types independently (i.e., a 4 parameter model), which tests the hypothesis that sponges affect both the attack rate and handling times of the prey; (2) a single estimate of attack rate, but separate estimates of handling times (for fouled and non-fouled) (3 parameter model), which test the hypothesis that sponges affect the interaction strength by facilitating crab feeding rates; (3) a model that permitted separate estimates of attack rates (for fouled and non-fouled), but only a single estimate of handling time (3 parameter model), which test the hypothesis that sponges change the likelihood of attack by crabs, and (4) a completely random model that fits only a single estimate of attack rate and handling time (2 parameter model), which serves as our null model. We did not directly measure attack rate or handling time, consequently these parameters were completely estimated from the model. Model fits and inferences about the effects of boring sponges on the interaction between crabs and oysters were made based on sample size-corrected Akaike Information Criterion (AICc).

Results

There was similar support for models 1 and 2 (Table 1), which is interesting given that both of these models allow separate estimates of handling times for crabs eating oysters with and without sponges. This may suggest that sponges are having the largest impacts on crab handling times, which is consistent with previous work indicating that boring sponges weakened mollusks shells (Duckworth & Peterson, 2013). Indeed, handling times (and therefore maximum consumption rates) were approximately 280% longer according to model 1 and 180% longer according to model 2 for crabs eating oysters without sponges relative to oysters with sponges (Table 1). However, the most supported model (model 1 in Table 1) also includes separate estimates of attack rates on oysters with and without boring sponges. While there is no evidence that Cliona spp. are unpalatable (Guida, 1976), lower attack rates on fouled oysters may suggest stone crabs have a higher propensity to attack and consume unfouled oysters. Regardless, the differences in attack rates were offset by longer handling times resulting in overall higher consumption of oysters in sponge treatments than in no sponge treatments (Fig. 1).

Table 1 Maximum likelihood results.

AICc values for each model. Estimates are presented for all parameters (α = attack rate and h = handling time) allowed to vary by treatment in a model (95% confidence intervals are presented underneath each estimate). With few observations (nobs = 38), corrected AIC (AICc) was used instead of AIC.

Model	Parameters	dAICc	df	Weight	No sponge	α	Sponge	No sponge	h	Sponge	
1	a*h	0	4	0.597	4.079 (1.457, 6.701)		1.970 (−3.416, 7.357)	0.112 (0.075, 0.149)		0.039 (−0.043, 0.123)	
2	h	1.1	3	0.352		2.534 (1.660, 3.408)		0.093 (0.063, 0.123)		0.051 (−0.011, 0.113)	
3	a	5.5	2	0.038	2.414 (1.140, 3.688)		1.97 (0.060, 5.353)		0.072 (0.048, 0.095)		
4	1	7.7	3	0.013		2.62 (1.690, 3.551)			0.074 (0.052, 0.096)		

Figure 1 Prey consumed over 24 h.

Amount of prey consumed by predators over a 24 h period using five increasing densities. Lines represent oysters (Crassostrea virginica) with sponges (Cliona spp.) (black) and oysters without sponges (gray), with standard error bars for each point (n = 4 trials). Attack rates and handling times used for each line were obtained from model 1 (see Table 1).

Discussion

We investigated how a fouling species that has expanded its range may be indirectly impacting the eastern oyster. Our results show that the presence of fouling from boring sponges will make oysters more susceptible to predation by crabs and likely other shell-crushing predators. One potential mechanism that we present here is a decrease in predator handling time for oysters with sponges compared to those without sponges. Indeed, boring sponges (Cliona celata) have been shown to weaken scallop shells by as much as 28% (Duckworth & Peterson, 2013). However, other studies have suggested that infestation by boring sponges did not impact stone crab handling times (Coleman, 2014). However, this difference may due in part to difference in the sizes of the stone crabs used in the two studies (mean carapace from Coleman 2014 =98.5 and current study 64.1 mm), such that defenses in shell strength were only evident for smaller crabs.

Understanding the potential effects of changes in species ranges and interactions are especially important for foundation species that provide structure that serves as primary habitat for communities of other species (Dayton, 1973). Specifically, global environmental change can facilitate species range expansions and alter local trophic interactions (Walther et al., 2002), which is critical information for mitigating and managing affected ecosystems. Boring sponges, Cliona spp., are experiencing range expansions potentially as a result of increased salinization of some estuarine ecosystems (Dunn, Eggleston & Lindquist, 2014; Lindquist, 2011). Boring sponge are generally found in areas with >15 ppt salinity (Carver, Thériault & Mallet, 2010; Hopkins, 1962; Lindquist, 2011), and they are being documented in increasing abundances further up-estuary as sea level and salinity rise (Hong & Shen, 2012; Lindquist, 2011).

Therefore, changes in the range of a fouling species in response to changes in habitat characteristics or climate change can have indirect consequences on the trophic interactions between important species (such as oysters and corals) and their natural enemies. Indeed, studies have shown that boring sponges are not impacted by high water temperatures or decreases in pH (due to an increase in atmospheric CO2) but instead these factors increase shell boring rates (Duckworth & Peterson, 2013). In addition, documented increases in the stone crab’s northward expansion along with boring sponges could enhance potential negative impacts to oyster fisheries in states such as Virginia and North Carolina. Overall, understanding how increases in predation risk as a result of fouling by boring sponges works in concert with other effects of global climate change (i.e., sea level rise, ocean acidification and increasing salinity) will have important implications for managing foundation species and the services they provide through fisheries, coastal protection, and ecosystem engineering.

This study highlights the need to consider how indirect biotic interactions can alter the interaction strengths between predators and prey. Indeed, boring sponges alone have modest impacts on oyster fitness. However, changes in the distribution and abundance of sponges, increases in boring efficiency, and the interactions between boring sponge and other species can lead to strong negative impacts on oysters and oyster reef communities. Such context dependent and indirect effects must be considered in future restoration and management aimed at recovering already heavily damaged oyster reef ecosystems (Beck et al., 2011; D’Anna, 2016).

Supplemental Information

Supplemental Information 1 Supplemental Document

Data and code used to generate figure and obtain model results.

Click here for additional data file.

Supplemental Information 2 R Code

R code used for statistics.

Click here for additional data file.

Data S1 Raw Data

Raw data collected from the experimental trials. Columns include: trial number, stone crab unique ID, the number of oysters, whether or not the oysters had sponges, data collection time points, and the number of oysters eaten.

Click here for additional data file.

We would like to thank Brian Silliman, the members of the Silliman Lab at the Duke Marine Lab, and lab assistants Thomas Guryan and Erin Tomaras for providing lab space and experimental assistance. In addition, we would like to thank David Kimmel, April Blakeslee, Krista McCoy, and the McCoy Labs for constructive comments on this research.

Additional Information and Declarations

Competing Interests

Author Contributions

Field Study Permissions

The authors declare there are no competing interests.

Cori J. Speights conceived and designed the experiments, performed the experiments, analyzed the data, wrote the paper, prepared figures and/or tables, reviewed drafts of the paper.

Michael W. McCoy conceived and designed the experiments, analyzed the data, contributed reagents/materials/analysis tools, wrote the paper, reviewed drafts of the paper.

The following information was supplied relating to field study approvals (i.e., approving body and any reference numbers):

For the collection of crabs, we had a collection permit approved by the North Carolina Division of Marine Fisheries. Field study approval number: 706671.

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
