# Peer review of "Range expansion of a fouling species indirectly impacts local species interactions"

_PeerJ, doi:10.7717/peerj.3911_

## Round 0.1 · original submission · Minor Revisions

In this manuscript the authors report the results of laboratory experiments to determine the effect of shell-boring sponges (Cliona spp.) on oyster predation by stone crabs. They find that oysters (Crassostrea virginica) whose shells have been compromised by boring sponges had much lower handling times, and higher rates of predation than oysters whose shells had not been bored. The manuscript is succinct and to the point, and all three external reviewers found the manuscript a worthwhile contribution that should be accepted pending minor revisions. I agree with their assessment. The comments of the reviewers are clear, thoughtful, and constructive, and don't need to be rehashed here. They should be addressed point-by-point in the cover letter for the revised manuscript. I will send a copy of the Word version of the manuscript with my comments and suggested editorial changes

Reviewer 1 ·

Basic reporting

The authors presented a well-written manuscript that meets the standards provided for basic reporting.

Experimental design

The research question is well defined and relevant, and the manuscript clearly states how this research fills a knowledge gap. There are some details that could be included in the methods, otherwise, the methods are sound.

Validity of the findings

The data are statistically sound. The conclusions and discussion are well stated, although the discussion could include more experiment specific comparisons to the literature.

Additional comments

The authors present an interesting manuscript examining the role that the boring sponge plays in in oyster predation by crabs. Specifically, they investigate how the presence of absence of the boring sponge affects attach rate and handling time of the stone crab, and important oyster predator. Although the experiment was fairly straightforward, I think this manuscript includes important information on this three species interaction. There are a few grammatical issues, and I think the intro/discussion can be expanded. However, I feel this issues should be easily addressed by the authors, and thus recommend publication after minor revisions. More specific details below:

Introduction: In general, the introduction does a good job putting the research in the broader context, however, I feel as though there are some specifics missing. For instance, why use the stone crab? That is, many studies focus on blue crabs and mud crabs as oyster predators. In fact, a number of recent studies have indicated that mud crabs are perhaps the most important oyster predator (Rindone and Eggleston 2011, Johnson et al. 2014, Carroll et al. 2015). Interestingly, the stone crab itself is also expanding its range. I don’t have problems with using the stone crab, but this species should be introduced in a little more detail. Likewise, for the boring sponge, a source showing that fouled individuals are more susceptible to predation would be useful.

P 3 LN 35-38: Two small issues. First, it is not entirely clear to me what this sentence is saying, so it probably needs to be reworded. Second, place (1983) after Schmitt et al. on line 36, it does not need to be placed at the end of the sentence.

P 4 LN 57-59: Be careful here – this study specifically investigates how the presence of sponges impact the trophic interaction between oysters and stone crabs (Line 60-62), but not how changes in the distributions of sponges are impacting this trophic relationship. This sentence is fine as justification, but this is not being investigated by this study.

P 4 LN 60: Here and throughout, make sure this is in past tense – “we investigate” should be “we investigated”

P 4 LN 65: Insert “than” between “survival” and “mollusks”.

P 5 LN 81-83: Were the crabs all the same sex?

P 5 LN 86: “Oysters of similar sizes were used…” What was the size range of oysters used?

P 5 LN 88-89: If the 5 hour and 24 hour oyster consumption checks were not used in the overall analysis (which it does not appear to be), this sentence can probably be omitted.

Page 6-7 LN 119-129: Make sure the model numbers here are the same as they are described in the results and the table. In the text, model 2 is separate estimates of attack rate, whereas in Table 1 and the results, model 2 is separate estimates of handling time.

Results: The authors should delve a little more into the results for model 1, since that was the best fit and estimated different attack rates for the different oyster groups as well. While I agree that the similar support of separate estimates of handling time model (see above comment) supports that handling time is probably most important, the best model also has very different attack rates. This should be mentioned.

Discussion: Should start narrow, then expand to broader implications. Summarize the results of the study first, and compare to the literature. Were the rates observed here comparable to other studies for non-fouled oysters? Which other studies of fouled oysters (or other organisms) also showed higher predation on the fouled organisms? The discussion is ok otherwise, but I think needs to include some oyster- and sponge-specific discussion before getting into all the broader implications. Further, I think the authors should also talk about the stone crab as a range expander – it has certainly expanded its range in NC, and since boring sponges are present all along the east coast, expanding stone crabs could be problematic for oysters in Virginia, for example, even if the sponge is not in new areas there.

·

Basic reporting

see below

Experimental design

see below

Validity of the findings

see below

Additional comments

July 31, 2017

PeerJ Editor

Re: Manuscript #18935
Review of Speights and McCoy “Range expansion of a fouling species indirectly impacts local species interactions”

Speights and McCoy used a series of laboratory trials to assess the effect of boring sponge infestation on predator-prey interactions between eastern oysters and stone crabs. In general, I thought the ms was clearly written and largely straightforward. The results of potentially longer handling times on non-infested oysters are certainly interesting, and could be of wide interest.

One overarching comment is that the manuscript is extremely succinct – at time perhaps too much so. For instance, the manuscript notes that there may be changes in the distribution of boring sponge, but only introduces stone crabs as an “important native” (e.g., Ln 61). In fact, stone crabs themselves may be experiencing a change in distribution (i.e., poleward range expansion) that is equally important in the context of this 3-member interaction – particularly in the region in which animals were collected for this study (e.g., Rindone and Eggleston 2011). This point could be expanded on in a revised manuscript (e.g., Ln 147).

Similarly, Sara Coleman (UNC-CH) published a thesis in 2014 entitled “The Effects of Boring Sponge on Oyster Soft Tissue, Shell Integrity, and Predator-Related Mortality”. Her results suggested that based on tissue content vs shell integrity, crabs should attack infested oysters over non-infested oysters. However, in feeding trials similar to Speights and McCoy, she found no difference in attack rates on infested vs. non-infested oysters. She concluded that the crushing strength of the crabs she used was so great (based on their size and reported crush force), that no oyster shell was a functional barrier to predation. Perhaps the size of stone crabs used by Speights and McCoy were smaller than those of Coleman – either way, it seems like some broader commentary on how these studies/results fit together could strengthen the discussion.

Other comments:

Ln 12: Could delete “In this study”, and just start “We investigate…”

Ln 37: I wondered how barnacle fouling on their shells increased the proximity of snails to predators. Since this is not intuitive, it would be good to detail the mechanism by which this occurs.

Ln 57 (and elsewhere): I understand I’m being perhaps a bit of a stickler, but I encourage the authors to be careful in how they present “recent changes in the distribution” of sponges. To my knowledge, neither Lindquist 2011 or Dunn et al. 2014 actually tested this. Lindquist mapped the present day distribution of sponges, and Dunn et al. examined sponge infestation in various substrates. Both worked across salinity gradients, and both certainly commented on the potential for sponges to be farther up-estuary (indeed, it’s a logical presumption). However, I’m not sure either had the long-term data to rigorously evaluate a shift in distribution.

It’s also entirely possible that I’ve missed some piece of evidence those researcher did provide, and I am also unfamiliar with the work of Hong and Shen 2012. I suppose I’m just asking the authors to be certain that they are not overstating what researchers may believe is happening, versus what researchers have actually documented.

Ln 86: “Oysters of similar sizes…” is vague. Provide actual sizes (and if sizes changed between trials for an individual crab, or across the four cycles [Ln 103]).

Ln 90-104: I thought the authors did a good job of accounting for the issues related to reusing crabs. I have no problem with interpreting their results keeping in mind that reuse occurred.

Ln 133-139: The increased handling time for non-infested oysters is the crux of this paper. However, I would like to see more detail on how handling time and attack rate (abort rates, etc.) were documented. Were these factors actually observed/recorded to feed Eqs 1&2? Or are they just estimated from the model fits of mortality. If it’s the latter, I have to wonder what sort of confidence intervals have to accompany those estimates. I think this is another place where the manuscript might benefit from further text/explanation.

Ln 151: “inland into estuaries” could be “up-estuary”

Good luck moving forward with this ms, yours,
[this is a signed review]

Joel Fodrie
Institute of Marine Sciences &
Department of Marine Sciences
University of North Carolina at Chapel Hill

·

Basic reporting

Line 65- please add “than” before survival

Experimental design

There are a couple of changes that should be made to ensure this work can be replicated.

Please provide the temperature and salinity of experimental tanks.

Figure 1- please include n=number of trials for each mean/standard error.

Figure 1- Please explain which equation was used for the lines shown here (I believe it was model 1 from your R code).

Validity of the findings

No comment.

Additional comments

This article is well-written. I was able to replicate all results from the code and data provided- nice job. The use of AIC to interpret the finding of this study was compelling. I am choosing minor revisions just because I think the temperature and salinity are necessary to fully interpret your results and compare them to other systems.

---

## Round 0.2 · accepted · Accept

The revised manuscript does a good job of addressing the reviewers concerns, and is now, in my opinion, ready for publication. I have made one last stab at further clarifying the sentence that runs from lines 33-37 in the attached PDF which you should incorporate in the production phase. This confused me and a couple of the reviewers.